# Dynamic Compression in Distributed Communications for Reduction of Transmission Losses

## Abstract

Recent trends in machine learning demonstrate an increasing demand for computational resources, compelling the ML community to leverage multiple devices for training. This concept is realized through distributed and federated learning approaches. However, despite the numerous advantages offered by these paradigms, they suffer from a significant limitation: the necessity of frequent information exchange between devices. A common solution to this issue involves compression. Existing operator definitions account only for second-moment deviation, which does not fully reveal the changes. Also, it is not assumed to vary during the learning process, but the forwarded information may have patterns, for example decreasing from iteration number. To address this limitation, we propose several novel classes of compression operators. Additionally, we introduce dynamic data types that adapt to the nature of the transmitted data. Our comprehensive theoretical analysis demonstrates the efficiency of this approach when applied to state-of-the-art algorithms, such as EF21, DIANA and DASHA. Experimental results further validate that the reduction in compression error and accelerates the convergence.

## 1 Introduction

Nonconvex optimization problems are ubiquitous in modern applications, especially in machine learning (Goodfellow et al., 2016; Jain et al., 2017; Schaipp et al., 2025). These tasks often involve massive datasets (Kocheturov et al., 2019) and models with millions of parameters, making training computationally expensive. Addressing such complexity requires specialized architectures that can effectively utilize both computational and memory resources.

A widely adopted solution is distributed optimization, which harnesses multiple interconnected devices to perform parallel computations and accelerate data access (Verbraeken et al., 2020; Kairouz et al., 2021). In this paradigm, each worker processes a subset of the data and communicates local updates to a central server, which aggregates them into a global model.

Unlike classical optimization, distributed one must balance not only computational efficiency but also communication efficiency. The latter is measured by the number of communication rounds and the size of transmitted messages. Since bandwidth-limited links often dominate the cost, communication can quickly become the main bottleneck.

A common strategy to alleviate this issue is *data compression* (Beznosikov et al., 2024b), which reduces the volume of transmitted information. Compression can be achieved through techniques such as low-rank decomposition (Yu et al., 2017), greedy sparsification (Shi et al., 2019), and quantization (Zhou et al., 2018). In this paper, we focus on data compression as a key tool for improving the efficiency of distributed optimization.

### 1.1 Compression

Compression operators have been utilized for an extended period and are extensively employed in distributed algorithms. These operators facilitate the transmission of reduced information while maintaining accuracy, a property that is particularly advantageous in distributed learning algorithms. The absence of these operators often results in significantly slower convergence of the algorithms in terms of the number of information transmitted.

**Definition 1** (Unbiased operator). *We say that a map $\mathcal{Q} : \mathbb{R}^d \to \mathbb{R}^d$ is an* **unbiased compression operator***, if there exists a constant $\omega \geq 0$, such that*

$$\mathbb{E}\left[\mathcal{Q}(x)\right] = x, \quad \mathbb{E}\left[\|\mathcal{Q}(x) - x\|^2\right] \leqslant \omega\|x\|^2, \qquad \forall x \in \mathbb{R}^d. \tag{1}$$

**Definition 2** (Biased operator). *We say that a map $\mathcal{C} : \mathbb{R}^d \to \mathbb{R}^d$ is a* **biased compression operator***, if there exists a constant $0 < \alpha \leqslant 1$, such that*

$$\mathbb{E}\left[\|\mathcal{C}(x) - x\|^2\right] \leqslant (1 - \alpha)\|x\|^2, \qquad \forall x \in \mathbb{R}^d. \tag{2}$$

## 2 RELATED WORKS AND OUR CONTRIBUTION

Machine learning methods are now everywhere, but training them still takes a huge amount of time. Distributed optimization is one of the main ways to speed things up, since multiple devices can work in parallel. The challenge, however, is that this speedup is not linear: adding more devices does not mean training gets equally faster. The main reason is communication—devices constantly need to exchange information, and this quickly becomes the bottleneck.

One of the earliest ideas to reduce communication was coordinate descent (Nesterov, 2012), later adapted to distributed settings (Richtárik & Takáč, 2016). Soon after, researchers began using quantization to compress updates, for example in the work of (Seide et al., 2014). This opened the door to more flexible strategies, such as Top-$k$ sparsification (Alistarh et al., 2018) and adaptive random sparsification (Beznosikov et al., 2024a), which attempt to balance accuracy with reduced communication.

Because unbiased operators are easier to analyze, much of the early work focused there. For example, QSGD (Alistarh et al., 2017) established convergence guarantees for SGD with quantized vectors. Later, methods such as DIANA (Li & Richtárik, 2020), MARINA (Gorbunov et al., 2021), and DASHA (Tyurin & Richtárik, 2022) improved communication efficiency by combining compression with variance reduction. A useful property of these algorithms is that the error introduced by compression shrinks over time as the transmitted updates get smaller.

In practice, though, biased operators often perform better. They can converge faster (Horvath et al., 2022; Beznosikov et al., 2024a), provided that error feedback mechanisms are used to correct for accumulated mistakes. This principle underpins methods like EF21 (Richtárik et al., 2021), which combine biased compression with error correction to maintain reliability.

Another important observation for both error-compensating techniques and variance-reduced ones, is that, as training goes on, the updates being sent naturally get smaller. If we keep using the same fixed compression scheme, we end up wasting precision on numbers that barely change. A smarter approach is to adjust the compression periodically, keeping just enough accuracy to ensure convergence while avoiding unnecessary communication.

Still, most existing operator definitions overlook a basic reality: quantization is limited by finite grids. This means that, in practice, algorithms often stop in a neighborhood of the true optimum rather than reaching it exactly. The natural next step is to design compressors that evolve with the learning process instead of staying fixed.

This paper takes a step in that direction by combining better mathematical models with adaptive compression in practice. Our main contributions are:

• **New compression operator definitions.** We introduce refined definitions for both biased and unbiased operators that explicitly take into account *absolute error*. This is important because:

1. Low-bit quantization always introduces some absolute error, which existing definitions ignore

2. Error accumulation during training unless it is modeled properly.

These new definitions give us a more realistic framework for analyzing compressed optimization.

• **Dynamic data type.** Since absolute errors do not grow indefinitely, traditional methods usually converge to a neighborhood of the solution instead of the exact optimum. To address this, we propose dynamic data types -— compression schemes that adapt to the current state of the algorithm.

This way, we can maintain convergence guarantees while reducing communication, and even reach arbitrary accuracy by gradually increasing precision.

• **Integration into state-of-the-art algorithms.**

We test our idea on DIANA (Li & Richtárik, 2020), EF21 (Richtárik et al., 2021), and DASHA (Tyurin & Richtárik, 2022), which represent the state of the art in distributed optimization. These algorithms are particularly well suited since they work with both biased and unbiased operators, and (unlike MARINA (Gorbunov et al., 2021)) do not require sending full gradients. By adapting the data type according to iteration count and gradient norms, we make communication more efficient without changing the algorithmic core.

• **Experimental validation.** Finally, we validate our approach with extensive experiments on ResNet and logistic regression. The results show that dynamic data types reduce communication costs significantly, without sacrificing accuracy. We also compare different update strategies, and find that adapting the compression scheme consistently improves convergence speed.

## 3 MAIN PART

We are interested in solving the following optimization problem:

$$\min_{x \in \mathbb{R}^d} \left\{ f(x) = \frac{1}{n} \sum_{i=1}^{n} f_i(x) \right\}, \tag{3}$$

where $f_i : \mathbb{R}^d \to \mathbb{R}$ is a smooth function for all $i \in [n] = \{1, \ldots, n\}$. We assume that the problem is solved by $n$ computational machines, each having exclusive access to $f_i$. We consider the star topology of the network, where all nodes are connected to one server and only. Our goal is to find an $\varepsilon$-solution of equation 3: a point $x \in \mathbb{R}^d$, where $\|\nabla f(x)\| \leq \varepsilon$ or $f(x) - f^* \leq \varepsilon$, where $f^* \stackrel{\text{def}}{=} \inf_{x \in \mathbb{R}^d} f(x) > -\infty$.

As mentioned above, to mitigate the communication costs in distributed optimization, compressions are employed. At first, we define new type of operators, describing quantizations that are used in practice.

### 3.1 NEW DEFINITIONS

A practical limitation of many compression analyses is the implicit use of *infinite* precision grids. Real systems quantize onto finite grids, which induces absolute errors near zero and clipping at large magnitudes. To model this, we work with *finite-grid* roundings and capture their effect via operator classes with additive error terms.

**Definition 3** (Rounding Unbiased compressor)**.** *Let $\{a_i\}_{i=0}^{M} \in \mathbb{R}$, where $a_0 = 0$ be an increasing sequence of non-negative numbers. We say that a map $\mathcal{D}_U(x, a) : \mathbb{R}^d \to \mathbb{R}^d$ is a **Rounding Unbiased compression operator**, if $a_k \leqslant |x_i| \leqslant a_{k+1}$*

$$\mathcal{D}_U(x, a)_i \stackrel{\text{def}}{=} \begin{cases} \text{sign}(x_i)a_k & \text{with probability} \quad \frac{a_{k+1} - |x_i|}{a_{k+1} - a_k} \\ \text{sign}(x_i)a_{k+1} & \text{with probability} \quad \frac{|x_i| - a_k}{a_{k+1} - a_k} \end{cases}, \quad i \in [d]. \tag{4}$$

Assumption on the finite grid's nature is crucial here. This compressor does not satisfy Assumption 2, as constant $\omega \to \infty$ when $x \to 0$, as we take $x_k$ closer and closer to 0, the variance bound is still fixed to the rounding lattice parameters. Therefore, there is no uniform bound on the second moment. Similar effect can be shown for biased compressor:

**Definition 4** (Rounding Biased Compressor)**.** *Let $\{a_i\}_{i=1}^{M} \in \mathbb{R}$ be an increasing sequence of non-negative numbers. We say that a map $\mathcal{D}_B(x, a) : \mathbb{R}^d \to \mathbb{R}^d$ is a **Rounding Biased compression operator**, if*

$$\mathcal{D}_B(x, a)_i \stackrel{\text{def}}{=} \text{sign}(x_i) \arg\min_{t \in a} |t - |x_i||, \quad i \in [d]. \tag{5}$$

In order to close this gap we propose a novel definitions for rounding operators with finite grid. It will allow to build convergence analysis for compressors mentioned above.

**Lemma 1.** *Rounding Biased compression operator 4 belongs to $\mathbb{B}^3(\alpha, \epsilon)$ with*

$$\alpha = \max_{a_k > 0} \left( \frac{4a_k a_{k+1}}{(a_k + a_{k+1})^2} \right), \quad \epsilon = d \max \left( a_0^2; \frac{a_1^2}{4}\alpha \right).$$

**Lemma 2.** *Rounding Unbiased compression operator 3 belongs to $\mathbb{U}(\omega, \epsilon)$ with*

$$\omega = \frac{1}{4} \max_{a_k > 0} \left( \frac{a_k}{a_{k+1}} + \frac{a_{k+1}}{a_k} + 2 \right), \quad \epsilon = d \frac{a_1^2}{4\omega}.$$

In the case of a biased rounding operator, we need to consider two possible cases must be considered: $a_0 = 0$ and $a_0 \neq 0$. For the sake of generality, the value of $\epsilon$ is taken to be the maximum of the two cases.

Another key property of the considered compressors is their closure under composition. Since the primary goal is to minimize transmitted information, one effective strategy is to communicate only a subvector of the gradient rather than its full representation. This approach is implemented in compressors like RandK (which transmits K randomly selected coordinates) and TopK (which sends the K coordinates with the largest magnitudes). As demonstrated in Lemmas 6 and 8, our proposed rounding schemes remain compatible with sparsification techniques such as TopK and RandK. This compatibility enables a substantial reduction in message sizes while maintaining algorithmic efficiency.

We employ these compressors into the modern distributed optimization methods: EF21 (Richtárik et al., 2021), DIANA (Li & Richtárik, 2020) and (Tyurin & Richtárik, 2022). The first one is utilized with biased version, whereas last two with unbiased.

---

**Algorithm 1** EF21

---

1: **Input:** starting point $x^0 \in \mathbb{R}^d$; $g_i^0 = \mathcal{C}(\nabla f_i(x^0))$ for $i = 1, \ldots, n$ (known by nodes and the master); learning rate $\gamma > 0$; $g^0 = \frac{1}{n} \sum_{i=1}^n g_i^0$ (known by master)
2: **for** $k = 0, 1, 2, \ldots, K - 1$ **do**
3:     Server broadcast $x^k$ to all nodes
4:     **for all nodes** $i = 1, \ldots, n$ **in parallel do**
5:         Compress
$$\Delta_i^k = \mathcal{D}_B(\nabla f_i(x^k) - g_i^k, \mathcal{A})$$
6:         Send $\Delta_i^k$ to the master
7:         Update local state $g_i^{k+1} = g_i^k + \Delta_i^k$
8:     **end for**
9:     $g^{k+1} = g^k + \frac{1}{n} \sum_{i=1}^n \Delta_i^k$
10:     $x^{k+1} = x^k - \gamma g^{k+1}$
11: **end for**

---

---

**Algorithm 2** DIANA

---

1: **Input:** starting point $x^0 \in \mathbb{R}^d$; $\left\{ h_i^0 \right\}_{i=1}^n$; $h^0 = \frac{1}{n} \sum_{i=1}^n h_i^0$; learning rate $\gamma > 0$; parameter $\alpha > 0$
2: **for** $k = 0, 1, \ldots, K - 1$ **do**
3:     Server broadcast $x^k$ to all nodes
4:     **for** $i = 1, \ldots, n$ in parallel **do**
5:         Compress shifted local gradient
$$\Delta_i^k = \mathcal{D}_U \left( \nabla f_i(x^k) - h_i^k, \mathcal{A} \right)$$
6:         Send $\Delta_i^k$ to the master
7:         Update local shift $h_i^{k+1} = h_i^k + \alpha \Delta_i^k$
8:     **end for**
9:     $g^{k+1} = h^k + \frac{1}{n} \sum_{i=1}^n \Delta_i^k$
10:     $x^{k+1} = x^k - \gamma g^{k+1}$
11:     $h^{k+1} = h^k + \alpha \frac{1}{n} \sum_{i=1}^n \Delta_i^k$
12: **end for**

---

---

**Algorithm 3** DASHA

---

1: **Input:** starting point $x^0 \in \mathbb{R}^d$; learning rate $\gamma > 0$, momentum $m \in (0, 1]$;
2: **for** $k = 0, 1, \ldots, K - 1$ **do**
3:      Server broadcast $x^k$ to all nodes
4:      **for** $i = 1, \ldots, n$ in parallel **do**
5:          Compress shifted local gradient
$$\Delta_i^k = \mathcal{D}_U \left( \nabla f_i(x^k) - \nabla f_i(x^{k-1}) - m \left( g_i^{k-1} - \nabla f_i(x^{k-1}) \right), \mathcal{A} \right)$$
6:          Update local state $g_i^{k+1} = g_i^k + \Delta_i^k$
7:          Send $\Delta_i^k$ to the master
8:      **end for**
9:      $g^{k+1} = g^k + \frac{1}{n} \sum_{i=1}^n \Delta_i^k$
10:     $x^{k+1} = x^k - \gamma g^{k+1}$
11: **end for**

---

Let us analyze the convergence results obtained by our methods. First, we introduce some important and common assumptions about the general form of the problem. These assumptions are crucial for understanding the theoretical underpinnings of the convergence analyses provided in the following theorems.

**Assumption 1** (Smoothness). *Every $f_i$ has $L_i$-Lipschitz gradient, i.e. for all $x, y \in \mathbb{R}^d$:*
$$\|\nabla f_i(x) - \nabla f_i(y)\| \leqslant L_i \|x - y\|.$$
*Moreover, we define $L^2 = \frac{1}{n} \sum_{i=1}^n L_i^2$.*

**Assumption 2** (Polyak-Lojasiewicz condition). *There exists $\mu > 0$ such that for all $x \in \mathbb{R}^d$:*
$$f(x) - f^* \leqslant \frac{1}{2\mu} \|\nabla f(x)\|^2.$$

**Convergence per iteration**

Convergence of all of the methods, described above, is based on Lyapunov function $V^k$ examination. Though, it might differ from algorithm to algorithm, both nonconvex and PL case. Both result are founded on the descent lemmas, that describe the convergence per iteration. Under given assumptions corresponding we can derive corresponding statements for inexact compressors:

**Lemma 3.** *Under Assumption 1 with $\gamma \sim \frac{1}{L}$ the progress per iteration is the following:*
$$V^{k+1} \leq V^k - \frac{\gamma}{2} \mathbb{E}\|\nabla f(x^k)\|^2 + \varepsilon(\mathcal{A})\gamma,$$
*where $V^k \sim f(x^k) - f^* + \frac{1}{n} \sum_{i=1}^n \|\nabla f_i(x^k) - g_i^k\|^2$*

**Lemma 4.** *Under Assumptions 1, 2 with asymptotically same $\gamma$ the progress per iteration is the following:*
$$V^{k+1} \leq (1 - \gamma\mu)V^k + \varepsilon(\mathcal{A})\gamma,$$
*where $V^k \sim f(x^k) - f^* + \frac{1}{n} \sum_{i=1}^n \|\nabla f_i(x^k) - g_i^k\|^2$*

Since there is an error, that does not diminish trough iterations, we should develop the approach, that will deal with this drawback. As mentioned above, this can be done by updating the grid iteratively.

Further we formalize and investigate the dynamic data type and its usage in the distributed optimization. These ideas will be implemented in the state-of-the-art algorithms, that use inexact unbiased and biased compressors.

## 3.2 DYNAMIC DATA TYPE

As the learning of the neural network progresses, the current state tend to converge to the local optimum. Therefore, the volumes of the transmitted information become less and less. Especially, this is frequently met in the case of channeling local gradients to the server. If the information is compressed by rounding, there is no need in preserving excessively large values all the way through the optimization. Hence, we will modify the rounding grid $\mathcal{A}$ as the learning progresses. Not only

this will reduce the amount of information transferred, but also better describe the data, since the grid will adapt to the current iteration.

While the operator's alterations are known at each compression, the discrepancy from the true value is likely to be non-zero. This discrepancy can accumulate from iteration to iteration, significantly impacting the convergence of algorithms.

Note that the grid $\mathcal{A}$ may be unique for each of the computing units, or it may be one set for all. In our experiments and further theory, we choose to use one set for all. In the following, we give some examples of the choice of $\mathcal{A}$:

1. One of the simplest idea is to use a maximum of coordinate $x$ over a maximum of $\mathcal{A}$ ($a_M = \max_i(x_i)$), filling the set according to the following rule: $a_{i-1} = p^{-1}a_i$, where $p > 1$. In a large number of cases, the best option is to take $p = 2$, which makes it easier to encode information about the set.

2. Another example would be to use percentiles of $x$ as the set $\mathcal{A}$. This method has performed well on our experiments. Let us describe it: introduce the function $g(x, b) : \quad g(x, b)_i = b_i\%_0(x)$, where $|b| = |a|, \forall i : \quad 0 \leqslant b_i \leqslant 100$ and $b_i\%_0(x)$ return $b_i$ percentile of $x$. The distribution of numbers in $b$ can be any, we tested three ideas – exponential, logarithmic and linear distribution, the last one gave the best convergence.

Let us look at the problem of rounding numbers when storing them in the memory of a computing device from a general point of view. Imagine that we need to store numbers within some limits, let it be an interval $[-10^3; 10^3]$ and we have 8 bits for it. This implies that a maximum of $2^8$ intervals can be encoded, with at least one containing a number. Thus, the error due to machine rounding does not exceed 4. If, after a period of time, the number in question decreases in size and falls into the interval $[-10; 10]$, then it is possible to achieve an error not exceeding 0.04 by using a new data type with the same size. This represents our principal approach to reducing the error.

It is now necessary to consider the implementation of ideas. It is important to recall that floating-point numbers are stored on computing devices in a particular manner. In accordance with the standard IEEE 754, single-precision numbers are allocated 4 bytes of information, with 1 bit designated for the sign of the number, 8 bits for the exponent, and the remaining 23 bits for the mantissa. This can be conceptualised as the division of a numerical line into intervals, with each interval representing a unique value. The membership of any given interval is then encoded. The number of such intervals is contingent upon the memory allocated for storing the number. In the context of a dynamic data type, the intervals can be modified in accordance with the convergence of the numbers. This approach allows for the use of less memory, thereby enabling the achievement of similar convergence results. This concept is illustrated in the accompanying figure 1.

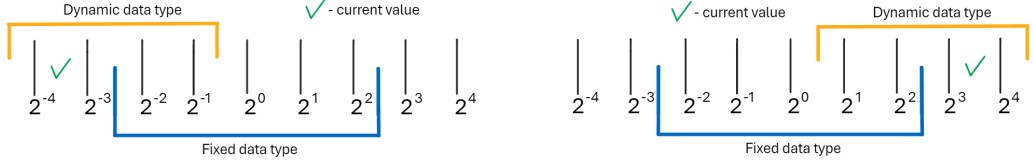

Figure 1: The implementation of the dynamic data type.

In this case, the use of a rounding compressor will result in the compressed numbers placed into a small-sized data type.

### 3.3 METHOD IMPLEMENTATION

Now consider how a dynamic data type is realized in the algorithm. We use the rounding operator $\mathcal{D}$ and the grid $\mathcal{A}$, which is known to the entire computational device. We also need a function that create $\mathcal{A}$ from the forwarding numbers, let us call it $set\_create$.

At some iteration steps of the algorithm, the conditions for changing the data type are checked. Such conditions can be either reaching of the forwarded information boundary by the norm, or the step number. If the condition is true, the set $\mathcal{A}$ is changed and sent to the devices.

We show below how these ideas apply to any algorithm in which the forwarded information tends to zero. Suppose that computational device $i$ transmits some information, denoted by $\mathcal{G}_i$, to the master. General view of the algorithm:

---

**Algorithm 4** Algorithm with dynamic data type

---
1: **Input:** algorithm parameters
2: Initialize grid $\mathcal{A}$
3: **for** $k = 0, 1, 2, \ldots, K - 1$ **do**
4:      **if** *Degree_cond is True or Iteration_cond is True* **then**
5:          Server change $\mathcal{A} = set\_create \left[ \mathcal{G} \stackrel{\text{def}}{=} \frac{1}{n} \sum_{i=1}^{n} \mathcal{G}_i \right]$ and sends $\mathcal{A}$ to all nodes
6:      **end if**
7:      Server and nodes run algorithm step, using $\mathcal{A}$ for send information from the $i$-th node
8: **end for**

---

Let us make some clarifications about *Degree_cond* and *Iteration_cond* used in the Algorithm 4.

We will proceed our explanation with the *Degree_cond*. This condition depends on the size of messages, received be the server. When these communications fall below a certain threshold, the grid is updated.

Now investigate *Iteration_cond*. In this context, it can be inferred that data type alterations are contingent upon the algorithmic step number or the number of steps elapsed since the previous data type modification. This approach necessitates more rigorous analysis to implement and may be contingent upon the specific problem at hand. Nevertheless, it remains applicable in certain instances. For example, if the rate at which the forwarded information $\mathcal{G}_i$ tends to zero is known, the number of iterations required to change the data type can be determined.

### 3.4 COMPREHEND CONVERGENCE ANALYSIS

After implementing the dynamic data type via changing compressors throughout the iterations, we might obtain the convergence to a better neighbourhood. We focus on approach, when we update the grid every fixed number of iterations

**Theorem 1** (Nonconvex rate with geometric grid refinement)**.** *Under Assumption 1 hold, choose* $\gamma \lesssim \frac{1}{L}$, *and update the grid every* $N$ *iterations so that* $\varepsilon(\mathcal{A}^{(s+1)}) \leq \rho \, \varepsilon(\mathcal{A}^{(s)})$ *with* $\rho \in (0, 1)$. *Then after* $K$ *iterations,*

$$\min_{0 \leq k < K} \mathbb{E} \|\nabla f(x^k)\|^2 \;\leq\; \frac{C_1}{K} \;+\; \frac{C_2 N}{K} \, \varepsilon(\mathcal{A}^{(0)}).$$

*Therefore, we yield the standard* $\mathcal{O}(1/K)$ *rate with no residual error.*

**Theorem 2.** *Under Assumptions 1,2, with asymptotically the same stepsize and geometric refinement schedule every* $N = \mathcal{O}\left(\frac{\log \rho}{\log(1 - \gamma\mu)}\right)$ *iterations, we obtain*

$$V^k \leq (1 - \gamma\mu)^{\frac{k}{2N}} V^0,$$

*which decays linearly to zero, where* $V^k \sim f(x^k) - f^* + \frac{1}{n} \sum_{i=1}^{n} \|\nabla f_i(x^k) - g_i^k\|^2$.

## 4 EXPERIMENTS

### 4.1 LOGISTIC REGRESSION

The datasets were taken from LibSVM (Chang & Lin, 2011) and were split into $n = 10$ equal parts, each associated with one of 10 clients. We first consider solving a logistic regression problem with a $L_2$ regularizer,

$$f(x) = \frac{1}{n} \sum_{i=1}^{n} \log \left(1 + \exp\left(-y_i a_i^\intercal x\right)\right) + \lambda \|x\|^2,$$

where $a_i \in \mathbb{R}^d, y_1 \in \{-1, 1\}$ are the training data and $\lambda > 0$ is regularizer parameter. We used $\lambda = 0.001$ in all experiments.

- **Dynamic data type.**

First experiments demonstrate the EF21 performance with dynamic data type. The rounding grid is chosen the following way: maximum element is chosen as the maximum norm of the first transmitted messages. Grid restructuring is conducted by dividing by $p = 2$ the moment the maximum element decreases at least in half. Bits stand for the memory, needed to store the lattice.

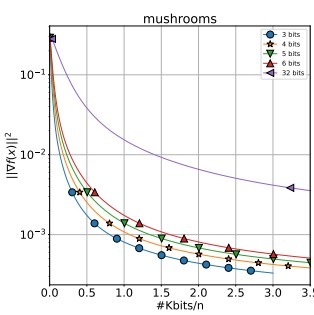 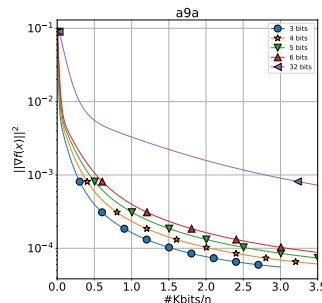 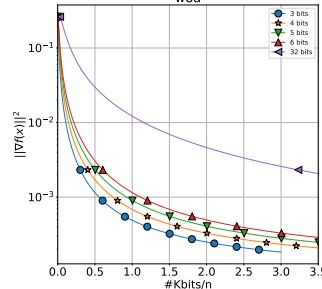

- **Comparison of static vs. dynamic data types**

Our next experiment carries with it the goal of showing how the convergence of the algorithms might change. Methods, classified as 1, are with static data type, whereas with 2, are for dynamic. $p$ stands for the base of rounding lattice reduction.

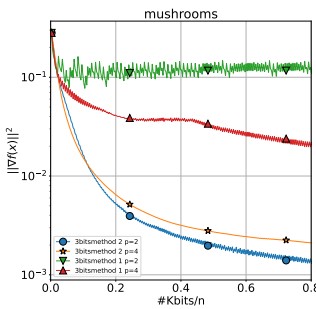 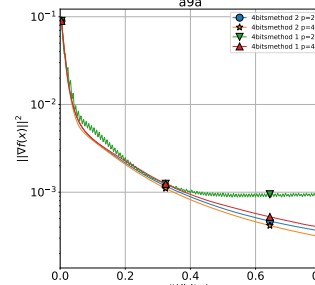 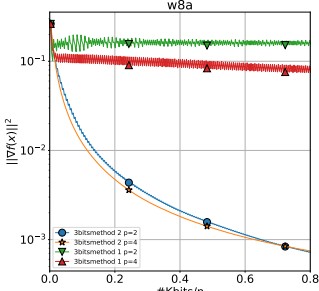

## 4.2 RESNET

Furthermore, analogous experiments were conducted on deep learning for image recognition using ResNet (He et al., 2016), which demonstrated the superiority of the dynamic data type over the static data type for the same size. The upper row displays convergence images of the dynamic type, while the lower row displays convergence images of the static type. $p$ is chosen as 2 for all the experiments.

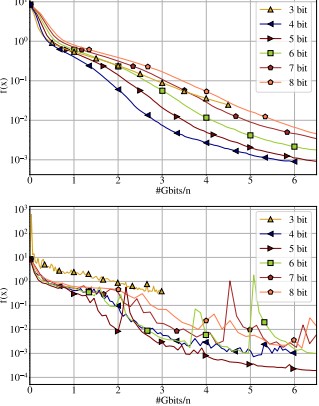 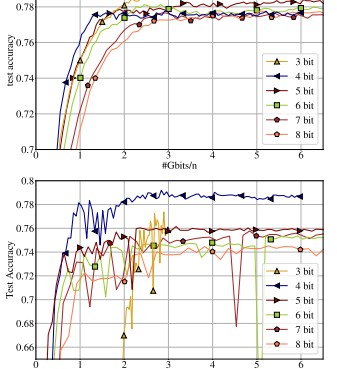 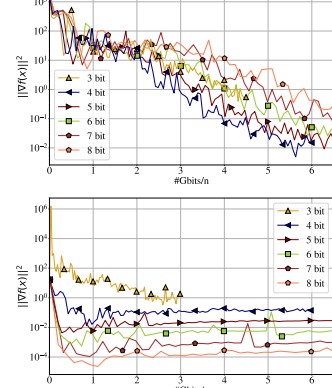

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

## A    COMPRESSORS ANALYSIS

In order to facilitate the analysis, it is necessary to introduce one further definition.

**Definition 5.** *Let $\alpha, \beta > 0, \epsilon_1, \epsilon_2 \geqslant 0$. We sat that $\mathcal{D} \in \mathbb{B}^1(\alpha, \beta, \epsilon_1, \epsilon_2)$ if*

$$\alpha \|x\|^2 \quad \leqslant \quad \mathbb{E}\left[\|\mathcal{D}(x)\|^2\right] + \epsilon_1 \quad \leqslant \quad \beta \langle \mathbb{E}\left[\mathcal{D}(x)\right] x\rangle + \epsilon_1 + \epsilon_2, \quad \forall x \in \mathbb{R}^d.$$

This definition permits a theoretical evaluation when two consecutive compression operators are applied. However, this class is not conducive to proof, so we shall instead demonstrate the theorem that connects $\mathbb{B}^1$ with $\mathbb{B}^3$.

**Lemma 5.** *If $\mathcal{C} \in \mathbb{B}^1(\alpha, \beta, \epsilon_1, \epsilon_2)$ and $\alpha \leqslant \beta^2$, then $\frac{1}{\beta}\mathcal{C} \in \mathbb{B}^3\left(\frac{\alpha}{\beta^2}, \frac{\epsilon_1 + 2\epsilon_2}{\beta^2}\right)$.*

*Proof.* Let $\lambda > 0$ be a scalar parameter.

$$
\begin{aligned}
\mathbb{E}\left[\|\lambda\mathcal{C}(x) - x\|^2\right] \quad &= \quad \mathbb{E}\left[\|\lambda\mathcal{C}(x)\|^2\right] - 2\langle\mathbb{E}\left[\lambda\mathcal{C}(x)\right], x\rangle + \|x\|^2 \\
&\leqslant \quad \lambda^2\left(\mathbb{E}\left[\|\mathcal{C}(x)\|^2\right] + \epsilon_1\right) - \lambda^2\epsilon_1 - 2\lambda\langle\mathbb{E}\left[\mathcal{C}(x)\right], x\rangle + \|x\|^2 \\
&\leqslant \quad \lambda^2\left(\beta\langle\mathbb{E}\left[\lambda\mathcal{C}(x)\right], x\rangle + \epsilon_1 + \epsilon_2\right) - 2\frac{\lambda}{\beta}\left(\beta\langle\mathbb{E}\left[\mathcal{C}(x)\right], x\rangle + \epsilon_1 + \epsilon_2\right) \\
&\quad + \|x\|^2 + 2\frac{\lambda}{\beta}\left(\epsilon_1 + \epsilon_2\right) - \lambda^2\epsilon_1 \\
&\leqslant \quad -\left(2\frac{\lambda}{\beta} - \lambda^2\right)\left(\beta\langle\mathbb{E}\left[\lambda\mathcal{C}(x)\right], x\rangle + \epsilon_1 + \epsilon_2\right) \\
&\quad + \|x\|^2 + 2\frac{\lambda}{\beta}\left(\epsilon_1 + \epsilon_2\right) - \lambda^2\epsilon_1.
\end{aligned}
$$

If we choose $\lambda \leqslant \frac{2}{\beta}$, then

$$-\left(2\frac{\lambda}{\beta} - \lambda^2\right)\left(\beta\langle\mathbb{E}\left[\lambda\mathcal{C}(x)\right], x\rangle + \epsilon_1 + \epsilon_2\right) + \|x\|^2 + 2\frac{\lambda}{\beta}\left(\epsilon_1 + \epsilon_2\right) - \lambda^2\epsilon_1$$

$$\leqslant -\alpha\left(2\frac{\lambda}{\beta} - \lambda^2\right)\|x\|^2 + \|x\|^2 + 2\frac{\lambda}{\beta}\left(\epsilon_1 + \epsilon_2\right) - \lambda^2\epsilon_1$$

$$= \left(1 - \alpha\left(2\frac{\lambda}{\beta} - \lambda^2\right)\right)\|x\|^2 + \frac{\epsilon_1 + 2\epsilon_2}{\beta^2}.$$

Minimize the multiplier of $\|x\|^2$ by $\lambda$, we obtain

$$\mathbb{E}\left[\|\lambda\mathcal{C}(x) - x\|^2\right] \quad \leqslant \quad \left(1 - \frac{\alpha}{\beta^2}\right)\|x\|^2 + \frac{\epsilon_1 + 2\epsilon_2}{\beta^2}.$$

$\square$

### A.1    ROUNDING COMPRESSION PARAMETERISATION

This section presents evidence that the constants in the estimates of the rounding operators 3 and 4 are accurate. The principal challenge in the estimation process is the segment $[0, \min(a_k > 0)]$, which is not obtained through methodologies analogous to those employed for the remainder of the set. Consequently, additional rounding errors are introduced.

It should be noted that the assumption is made that the maximal element in $x$ does not exceed the maximal element from set $\mathcal{A}$. This is achieved through the use of set problems and algorithms.

The constants $\zeta$ are evaluated in reference (Beznosikov et al., 2024a), and thus, the proof for them is omitted.

*Proof of lemma 2.* Let $|x_i| \in_{0, a_1} =_{a_0, a_1} \forall i \in \overline{1, d}$. In order to estimate $\epsilon$, it is necessary to identify the maximum value of $\mathbb{E}\left[\|\mathcal{D}(x)\|^2\right] - \zeta\|x\|^2$. For the sake of simplicity, we will assume that

$x_i \geqslant 0$.

$$\max_x \left( \mathbb{E}\left[\|\mathcal{D}(x)\|^2\right] - \zeta\|x\|^2 \right) = \max_x \left( \sum_{i=1}^{d} \left( \mathbb{E}\left[(\mathcal{D}(x_i))^2\right] - \zeta x_i^2 \right) \right)$$

$$= d \max_{x_i} \left( \mathbb{E}\left[(\mathcal{D}(x_i))^2\right] - \zeta x_i^2 \right). \tag{6}$$

The last equality is valid since the compression operator acts equally on each of the coordinates, and it is assumed that any input is possible.

$$\mathbb{E}\left[(\mathcal{D}(x_i))^2\right] = \begin{cases} 0, & \text{with probability } \frac{a_1 - x_i}{a_1} \\ a_1^2, & \text{with probability } \frac{x_i}{a_1} \end{cases} = x_i a_1.$$

Substituting this result into equation 6, we obtain

$$d \max_{x_i} \left( \mathbb{E}\left[(\mathcal{D}(x_i))^2\right] - \zeta x_i^2 \right) = d \max_{x_i} \left( x_i a_1 - \zeta x_i^2 \right) = d \frac{a_1^2}{4\zeta}.$$

$\square$

*Proof of lemma 1.* In this proof, we consider two cases: when $a_0 = 0$ and $a_0 > 0$. For the sake of simplicity, we will assume that $x_i \geqslant 0$. In order to estimate $\epsilon$, it is necessary to identify the maximum value of $\mathbb{E}\left[\|\mathcal{D}(x) - x\|^2\right] - (1 - \zeta)\|x\|^2$.

- Let us look at case $a_0 = 0$.

$$\max_x \left( \mathbb{E}\left[\|\mathcal{D}(x) - x\|^2\right] - (1 - \zeta)\|x\|^2 \right)$$

$$= \max_x \sum_{i=1}^{d} \mathbb{E}\left[(\mathcal{D}(x_i) - x_i)^2\right] - (1 - \zeta)x_i^2$$

$$= d \max_{x_i} \left( \mathbb{E}\left[(\mathcal{D}(x_i) - x_i)^2\right] - (1 - \zeta)x_i^2 \right). \tag{7}$$

  The last equality is valid since the compression operator acts equally on each of the coordinates, and it is assumed that any input is possible.

$$\mathbb{E}\left[(\mathcal{D}(x_i) - x_i)^2\right] = \begin{cases} x_i^2, & x_i < \frac{a_1}{2} \\ (a_1 - x_i)^2, & x_i \geqslant \frac{a_1}{2} \end{cases}.$$

  Substituting this result into equation 7, we obtain

$$d \max_{x_i} \left( \mathbb{E}\left[(\mathcal{D}(x_i) - x_i)^2\right] - (1 - \zeta)x_i^2 \right) = d \max \begin{cases} \zeta x_i^2, & x_i < \frac{a_1}{2} \\ a_1^2 - 2a_1 x_i + \zeta x_i^2, & x_i \geqslant \frac{a_1}{2} \end{cases}$$

$$= \zeta \frac{a_1^2}{4}$$

- Look at case $a_0 > 0$. In this instance, the error can be more easily estimated by considering $x_i = 0 \ \forall i \in \overline{1, d}$. This is the point at which the largest rounding error occurs. Use equality equation 7 and obtain

$$\max_x \left( \mathbb{E}\left[\|\mathcal{D}(x) - x\|^2\right] - (1 - \zeta)\|x\|^2 \right)$$

$$= d \max_{x_i} \left( \mathbb{E}\left[(\mathcal{D}(x_i) - x_i)^2\right] - (1 - \zeta)x_i^2 \right)$$

$$= d a_0.$$

Generalisation of the result can be written as $d \max \left( a_0^2; \ \zeta \frac{a_1^2}{4} \right)$ $\square$

## A.2 UNBIASED ROUNDING COMPRESSION

We will now proceed to a general analysis of the application of two compression operators in a row. It is essential to consider the error introduced by compression of the forwarded information and the resulting reduction in convergence speed. For the sake of clarity, it is advisable to consider the cases of biased and unbiased rounding compression separately.

**Lemma 6.** *Let $\mathcal{D}_U \in \mathbb{U}(\zeta_0, \epsilon_0)$ and $\mathcal{Q} \in \mathbb{U}(\zeta_1, \epsilon_1)$, then $\mathcal{D}_U(\mathcal{Q})$ belongs to $\mathbb{U}(\zeta_0\zeta_1, \epsilon_0 + \epsilon_1)$.*

*Proof.*
$$\left\| \mathcal{D}_U(\mathcal{Q}(x)) \right\|^2 \quad \leqslant \quad \zeta_0 \left\| \mathcal{Q}(x) \right\|^2 + \epsilon_0 \quad \leqslant \quad \zeta_0\zeta_1 \left\| x \right\|^2 + \epsilon_0 + \epsilon_1.$$
$\square$

**Lemma 7.** *Let $\mathcal{D}_U \in \mathbb{U}(\zeta_0, \epsilon_0)$ and $\mathcal{C} \in \mathbb{B}^1(\alpha, \beta, \epsilon_1, \epsilon_2)$, then $\frac{1}{\zeta_0\beta}\mathcal{D}_U(\mathcal{C})$ belongs to $\mathbb{B}^3\left( \frac{\alpha}{(\zeta_0\beta)^2}, \frac{\epsilon_1 + 2(\epsilon_0 + \zeta_0\epsilon_2)}{(\zeta_0\beta)^2} \right).$*

*Proof.* Let us first show that $\mathcal{D}_U(\mathcal{C})$ belongs to $\mathbb{B}^1(\alpha, \zeta_0\beta, \epsilon_1, \epsilon_0 + \zeta_0\epsilon_2)$. For this note
$$\left\| x \right\|^2 \quad \leqslant \quad \mathbb{E}\left[ \left\| \mathcal{D}_U(x) \right\|^2 \right] \quad \forall x \in \mathbb{R}^d.$$
Keep it in mind, we obtain
$$\alpha \left\| x \right\|^2 \quad \leqslant \quad \mathbb{E}\left[ \left\| \mathcal{C}(x) \right\|^2 \right] + \epsilon_1 \quad \leqslant \quad \mathbb{E}\left[ \left\| \mathcal{D}_U(\mathcal{C}(x)) \right\|^2 \right] + \epsilon_1 \quad \leqslant \quad \zeta_0 \left\| \mathcal{C}(x) \right\|^2 + \epsilon_0 + \epsilon_1.$$
Taking one more expectation, we get:
$$\begin{aligned} \alpha \left\| x \right\|^2 \quad &\leqslant \quad \mathbb{E}\left[ \left\| \mathcal{D}_U(\mathcal{C}(x)) \right\|^2 \right] + \epsilon_1 \quad \leqslant \quad \zeta_0 \mathbb{E}\left[ \left\| \mathcal{C}(x) \right\|^2 \right] + \epsilon_0 + \epsilon_1 \\ &\leqslant \quad \zeta_0\beta \langle \mathbb{E}\left[ \mathcal{C}(x) \right], x \rangle + \epsilon_0 + \epsilon_1 + \zeta_0\epsilon_2 \\ &= \quad \zeta_0\beta \langle \mathbb{E}\left[ \mathcal{D}_U(\mathcal{C}(x)) \right], x \rangle + \epsilon_0 + \epsilon_1 + \zeta_0\epsilon_2. \end{aligned}$$
After use lemma 5, we obtain the result.
$$\frac{1}{\zeta_0\beta}\mathcal{D}_U(\mathcal{C}) \in \mathbb{B}^3\left( \frac{\alpha}{(\zeta_0\beta)^2}, \frac{\epsilon_1 + 2(\epsilon_0 + \zeta_0\epsilon_2)}{(\zeta_0\beta)^2} \right).$$
$\square$

As can be observed, the application of the unbiased rounding operator results in the analysis indicating that the obtained operators are generally consistent with the previously introduced classes. Consequently, when two compressors are applied in this instance, convergence analysis is possible.

### A.3 Biased rounding compression

**Lemma 8.** *Let $\mathcal{D}_B \in \mathbb{B}^3(\zeta_0, \epsilon_0)$ and $\mathcal{Q} \in \mathbb{U}(\zeta_1, \epsilon_1)$, then $\frac{1}{\zeta_1}\mathcal{D}_B(\mathcal{Q})$ belongs to $\mathbb{B}^3\left( \zeta_0, \frac{\epsilon_0 + (1 - \zeta_0)\epsilon_1}{\zeta_1} \right).$*

*Proof.*
$$\begin{aligned} \frac{1}{\zeta_1} \left\| \mathcal{D}_B(\mathcal{Q}(x)) - x \right\|^2 \quad &\leqslant \quad \frac{1}{\zeta_1}(1 - \zeta_0) \left\| \mathcal{Q}(x) \right\|^2 + \frac{1}{\zeta_1}\epsilon_0 \\ &\leqslant \quad (1 - \zeta_0) \left\| x \right\|^2 + \frac{1}{\zeta_1} \left( \epsilon_0 + (1 - \zeta_0)\epsilon_1 \right). \end{aligned}$$
$\square$

**Lemma 9** (Counterexample). *$\exists \mathcal{C}_1 \in \mathbb{B}^3(\zeta_1, 0)$, $\mathcal{C}_2 \in \mathbb{B}^3(\zeta_2, 0)$ that $\nexists\zeta$ such that $\mathcal{C}_1 \circ \mathcal{C}_2 \in \mathbb{B}^3(\zeta, 0)$. We can write this statement as follows:*
$$\exists x \in \mathbb{R}^d, \exists \mathcal{C}_1, \mathcal{C}_2 \in \mathbb{B}^3(\cdot, 0) \quad : \quad \left\| \mathcal{C}_1(\mathcal{C}_2(x)) - x \right\|^2 > \left\| x \right\|^2.$$

*Proof.* Let us set $b_0 > 0 \in \mathbb{R}^1$ and take $x = b_0$. Denote
$$\begin{aligned} \mathcal{A}_2 \quad &= \quad (a_k)_{k \in \mathbb{Z}} \quad = \quad \left( \ldots, b_0\left( 2 - \sqrt{2 + 2\varepsilon} \right), b_0\sqrt{2 + \varepsilon}, \ldots \right), \quad 0 < \varepsilon \ll 1 \\ \mathcal{A}_1 \quad &= \quad (a_k)_{k \in \mathbb{Z}} \quad = \quad \left( \ldots, b\left( 2 - \sqrt{2 + 2\varepsilon} \right), b\sqrt{2 + \varepsilon}, \ldots \right), \quad b = b_0\sqrt{2 + \varepsilon} \end{aligned}$$
in such a way that $\inf \mathcal{A}_1 = \inf \mathcal{A}_2 = 0$ and $\sup \mathcal{A}_1 = \sup \mathcal{A}_2 = \infty$. Then says that $\mathcal{C}_2$ and $\mathcal{C}_1$ are **general biased rounding** with sets $\mathcal{A}_2$ and $\mathcal{A}_1$ respectively.

First of all, let us confirm that so defined compression operators satisfy the conditions on $\zeta = \sup_{k \in \mathbb{Z}} \frac{4 a_k a_{k+1}}{(a_k + a_{k+1})^2}$.

$$
\begin{aligned}
\zeta &= \sup_{k \in \mathbb{Z}} \frac{4 a_k a_{k+1}}{(a_k + a_{k+1})^2} = \sup_{k \in \mathbb{Z}} \frac{4 \frac{a_{k+1}}{a_k}}{\left(1 + \frac{a_{k+1}}{a_k}\right)^2} = \sup_{k \in \mathbb{Z}} \frac{4 \left(1 + \frac{a_{k+1} - a_k}{a_k}\right)}{\left(2 + \frac{a_{k+1} - a_k}{a_k}\right)^2} \\
&= \sup_{k \in \mathbb{Z}} \frac{4 \left(1 + \frac{a_{k+1} - a_k}{a_k}\right)}{4 \left(1 + \frac{a_{k+1} - a_k}{a_k}\right) + \left(\frac{a_{k+1} - a_k}{a_k}\right)^2} < 1.
\end{aligned}
$$

Now we show that $\mathcal{C}_2(b_0) = \sqrt{2 + \varepsilon} b_0 = b$ and $\mathcal{C}_1(b) = \sqrt{2 + \varepsilon} b = (2 + \varepsilon) b_0$. Write proof for $\mathcal{C}_2(x)$

$$
\begin{aligned}
\mathcal{C}_2(x) &= \mathcal{C}_2(b_0) \\
&= \operatorname{sign}(b_0) \underset{t \in (a_k)}{\arg\min} |t - |b_0|| \\
&= \operatorname{sign}(b_0) \operatorname{argmin} \left\{ \left| b_0 \left(2 - \sqrt{2 + 2\varepsilon}\right) - |b_0| \right|, \left| b_0 \sqrt{2 + \varepsilon} - |b_0| \right| \right\} \\
&= b_0 \operatorname{argmin} \left(\sqrt{2 + 2\varepsilon} - 1, \sqrt{2 + \varepsilon} - 1\right) \\
&= b_0 \sqrt{2 + \varepsilon}.
\end{aligned}
$$

Note that proof for $b \sqrt{2 + \varepsilon} = \mathcal{C}_1(b) = \mathcal{C}_1(\mathcal{C}_2(b_0)) = \mathcal{C}_1(\mathcal{C}_2(x))$ is similar.

$$
\begin{aligned}
\mathbb{E}\left[\|\mathcal{C}_1(\mathcal{C}_2(x)) - x\|^2\right] &= \|\mathcal{C}_1(\mathcal{C}_2(x)) - x\|^2 \\
&= \|\mathcal{C}_1(\mathcal{C}_2(b_0)) - b_0\|^2 = \|(2 + \varepsilon) b_0 - b_0\|^2 = (1 + \varepsilon)^2 \|b_0\|^2 \\
&= (1 + \varepsilon)^2 \|x\|^2 > \|x\|^2.
\end{aligned}
$$
$\square$

However, the overall analysis does not consider the interactions between compression operators, which may result in inaccurate estimates. Therefore, we provide additional estimates for TopK and RandK sparsification.

**Lemma 10.** *Let $\mathcal{C}$ be a TopK and $\mathcal{D}_B$ rounding biased operator 4 with $a_0 = 0$. Then*

$$
\|\mathcal{D}_B(\mathcal{C}(x)) - x\|^2 \leqslant \left(1 - \zeta \frac{k}{d}\right) \|x\|^2 + k \left(\frac{a_1}{2}\right)^2.
$$

*Proof.* Firs look at $\mathcal{D}_B(\mathcal{C}(x))$.

$$
\mathcal{D}_B(\mathcal{C}(x)) = \mathcal{D}_B\left(\sum_{i=1}^{k} x_{(i)}\right).
$$

$$
\begin{aligned}
\|\mathcal{D}_B(\mathcal{C}(x)) - x\|^2 &= \sum_{i=1}^{d} \left(\mathcal{D}_B(\mathcal{C}(x_i)) - x_i\right)^2 = \sum_{i=1}^{k} \left(\mathcal{D}_B(x_{(i)}) - x_{(i)}\right)^2 + \sum_{i=k+1}^{d} \left(x_{(i)}\right)^2 \\
&\leqslant (1 - \zeta) \sum_{i=1}^{k} \left(x_{(i)}\right)^2 + \sum_{i=k+1}^{d} \left(x_{(i)}\right)^2 + k \left(\frac{a_1}{2}\right)^2 \\
&= \|x\|^2 - \zeta \sum_{i=1}^{k} \left(x_{(i)}\right)^2 + k \left(\frac{a_1}{2}\right)^2 \\
&= \|x\|^2 - \zeta \frac{k}{d} \sum_{i=1}^{k} \frac{d}{k} \left(x_{(i)}\right)^2 + k \left(\frac{a_1}{2}\right)^2 \\
&\leqslant \left(1 - \zeta \frac{k}{d}\right) \|x\|^2 + k \frac{3 a_1^2}{4}
\end{aligned}
$$
$\square$

## B  PER ITERATION PROGRESS

**Lemma 11** (EF21 iteration). *Let Assumption 1, be satisfied and $\mathcal{C} \in \mathbb{B}^3 (\alpha, \varepsilon)$. Choose*

$$0 < \gamma \leq \frac{1}{L\left(1 + \sqrt{\frac{\beta}{\theta}}\right)},$$

*where $\theta = 1 - \sqrt{1-\alpha}, \beta = \frac{1-\alpha}{1-\sqrt{1-\alpha}}$. Define $V^t = \mathbb{E}\left[f(x^t) + \frac{\gamma}{2\theta} \frac{1}{n} \sum_{i=1}^{n} \|\nabla f_i(x^t) - g_i^t\|^2\right]$. Then, we have*

$$V^{t+1} \leq V^t - \frac{\gamma}{2}\mathbb{E}\|\nabla f(x^t)\|^2 + \frac{\gamma}{2\theta}\varepsilon,$$

*and*

$$\frac{1}{T}\sum_{t=0}^{T-1} \mathbb{E}\|\nabla f(x^t)\|^2 \leq \frac{2V^0}{\gamma T} + \frac{\varepsilon}{\theta}.$$

*Additionally, if Assumptions **??** and 2 are satisfied, $V^t = \mathbb{E}\left[f(x^t) - f^* + \frac{\gamma}{2\theta}\frac{1}{n}\sum_{i=1}^{n} \|\nabla f_i(x^t) - g_i^t\|^2\right]$ and $\gamma$ is chosen as*

$$0 < \gamma \leq \min\left\{\frac{1}{L(1 + \sqrt{\frac{2\beta}{\theta}})}, \frac{\theta}{2\mu}\right\}.$$

*Then,*

$$V^{t+1} \leq (1 - \gamma\mu)V^t + \frac{\gamma}{\theta}\varepsilon,$$

*and*

$$V^{t+1} \leq (1 - \gamma\mu)^{t+1}V^0 + \frac{\varepsilon}{\theta\mu}.$$

*Proof.* From $L$-smoothness we obtain

$$
\begin{aligned}
f(x^{t+1}) &\leq f(x^t) + \langle \nabla f(x^t), x^{t+1} - x^t\rangle + \frac{L}{2}\|x^{t+1} - x^t\|^2 \\
&\leq f(x^t) - \gamma\langle \nabla f(x^t), g^t\rangle + \frac{L\gamma^2}{2}\|g^t\|^2 \\
&= f(x^t) - \frac{\gamma}{2}\|\nabla f(x^t)\|^2 + \left(\frac{L\gamma^2}{2} - \frac{\gamma}{2}\right)\|g^t\|^2 + \frac{\gamma}{2}\|\nabla f(x^t) - g^t\|^2 \\
&= f(x^t) - \frac{\gamma}{2}\|\nabla f(x^t)\|^2 + \left(\frac{L\gamma^2}{2} - \frac{\gamma}{2}\right)\|g^t\|^2 + \frac{\gamma}{2}\left\|\frac{1}{n}\sum_{i=1}^{n}\nabla f_i(x^t) - g_i^t\right\|^2 \\
&\leq f(x^t) - \frac{\gamma}{2}\|\nabla f(x^t)\|^2 + \left(\frac{L\gamma^2}{2} - \frac{\gamma}{2}\right)\|g^t\|^2 + \frac{\gamma}{2}\frac{1}{n}\sum_{i=1}^{n}\left\|\nabla f_i(x^t) - g_i^t\right\|^2.
\end{aligned}
$$

Now we analyze the term $G^{t+1} = \frac{1}{n}\sum_{i=1}^{n}\|\nabla f_i(x^{t+1}) - g_i^{t+1}\|^2$. Taking the conditional expectation $\mathbb{E}_{\xi_t} = \mathbb{E}[\cdot \mid x^1, \ldots, x^t]$, we have

$$
\begin{aligned}
\mathbb{E}_{\xi_t}\frac{1}{n}\sum_{i=1}^{n}\left\|\nabla f_i(x^{t+1}) - g_i^{t+1^2}\right\|^2 &= \mathbb{E}_{\xi_t}\frac{1}{n}\sum_{i=1}^{n}\left\|\nabla f_i(x^{t+1}) - \mathcal{D}_B\left(\nabla f(x_i^{t+1}) - g_i^t\right) - g_i^t\right\|^2 \\
&\leq \mathbb{E}_{\xi_t}\frac{1-\alpha}{n}\sum_{i=1}^{n}\|\nabla f(x^{t+1}) - g_i^t\|^2 + \varepsilon \\
&\leq (1-\alpha)(1+s^{-1})L^2\mathbb{E}_{\xi_t}\|x^{t+1} - x^t\|^2 \\
&\quad + (1-\alpha)(1+s)\frac{1}{n}\sum_{i=1}^{n}\|\nabla f_i(x^t) - g_i^t\|^2 + \varepsilon
\end{aligned}
$$

Use the fact, that $x^{t+1} - x^t = g^t$, then

$$
\mathbb{E}_{\xi_t} \frac{1}{n} \sum_{i=1}^{n} \left\| \nabla f_i(x^{t+1}) - g_i^{t+1^2} \right\|^2 = (1-\alpha)(1+s^{-1}) L^2 \gamma^2 \mathbb{E}_{\xi_t} \|g^t\|^2
$$

$$
+ (1-\alpha)(1+s) \frac{1}{n} \sum_{i=1}^{n} \|\nabla f_i(x^t) - g_i^t\|^2 + \varepsilon
$$

Take $s = \frac{1}{\sqrt{1-\alpha}} - 1$, hence,

$$
\mathbb{E}_{\xi_t} G^{t+1} \leq (1-\theta) G^t + \beta L^2 \gamma^2 \mathbb{E}_{\xi_t} \|g^t\|^2 + \varepsilon,
$$

where $\theta = 1 - \sqrt{1-\alpha}$, $\beta = \frac{1-\alpha}{1-\sqrt{1-\alpha}}$.

To combine these bounds, we 1) add $G^{t+1}$ with a multiplier $\frac{\gamma}{2\theta}$ 2) take full expectation:

$$
\mathbb{E}\left[ f(x^{t+1}) + \frac{\gamma}{2\theta} G^{t+1} \right] \leq \mathbb{E}f(x^t) - \frac{\gamma}{2} \mathbb{E}\|\nabla f(x^t)\|^2 + \left( \frac{L\gamma^2}{2} - \frac{\gamma}{2} + \frac{\beta}{2\theta} L^2 \gamma^3 \right) \mathbb{E}\|g^t\|^2
$$

$$
+ \left( \frac{\gamma}{2} + \frac{\gamma}{2\theta}(1-\theta) \right) \mathbb{E}G^t + \frac{\gamma}{2\theta} \varepsilon
$$

$$
\leq \mathbb{E}f(x^t) - \frac{\gamma}{2} \mathbb{E}\|\nabla f(x^t)\|^2 + \left( \frac{L\gamma^2}{2} - \frac{\gamma}{2} + \frac{\beta}{2\theta} L^2 \gamma^3 \right) \mathbb{E}\|g^t\|^2
$$

$$
+ \frac{\gamma}{2\theta} \mathbb{E}G^t + \frac{\gamma}{2\theta} \varepsilon.
$$

After the choice of $\gamma$ we have

$$
\mathbb{E}\left[ f(x^{t+1}) + \frac{\gamma}{2\theta} G^{t+1} \right] \leq \mathbb{E}f(x^t) - \frac{\gamma}{2} \mathbb{E}\|\nabla f(x^t)\|^2 + \frac{\gamma}{2\theta} \mathbb{E}G^t + \frac{\gamma}{2\theta} \varepsilon.
$$

Therefore, we get

$$
\frac{1}{T} \sum_{t=1}^{T} \mathbb{E}\|\nabla f(x^t)\|^2 \leq \frac{2f(x^0) + \frac{\gamma}{\theta} G^0}{\gamma T} + \frac{\varepsilon}{\theta}.
$$

In PL case we add $G^{t+1}$ with multiplier $\frac{\gamma}{\theta}$ and extract $f^*$:

$$
\mathbb{E}\left[ f(x^{t+1}) - f^* + \frac{\gamma}{2\theta} G^{t+1} \right] \leq \mathbb{E}f(x^t) - f^* - \frac{\gamma}{2} \mathbb{E}\|\nabla f(x^t)\|^2
$$

$$
+ \left( \frac{L\gamma^2}{2} - \frac{\gamma}{2} + \frac{\beta}{\theta} L^2 \gamma^3 \right) \mathbb{E}\|g^t\|^2 + \left( \frac{\gamma}{2} + \frac{\gamma}{\theta}(1-\theta) \right) \mathbb{E}G^t + \frac{\gamma}{\theta} \varepsilon
$$

$$
\leq (1-\gamma\mu)(f(x^t) - f^*) + \left( \frac{L\gamma^2}{2} - \frac{\gamma}{2} + \frac{\beta}{\theta} L^2 \gamma^3 \right) \mathbb{E}\|g^t\|^2
$$

$$
+ \left( 1 - \frac{\theta}{2} \right) \frac{\gamma}{2\theta} \mathbb{E}G^t + \frac{\gamma}{\theta} \varepsilon.
$$

After the choice of $\gamma$ we have

$$
\mathbb{E}\left[ f(x^{t+1}) - f^* + \frac{\gamma}{2\theta} G^{t+1} \right] \leq (1-\gamma\mu) \mathbb{E}\left[ f(x^t) - f^* + \frac{\gamma}{2\theta} G^t \right] + \frac{\gamma}{\theta} \varepsilon.
$$

After unrolling the sequence we obtain

$$
\mathbb{E}\left[ f(x^{t+1}) - f^* + \frac{\gamma}{2\theta} G^{t+1} \right] \leq (1-\gamma\mu)^{t+1} \left[ f(x^0) - f^* + \frac{\gamma}{2\theta} G^0 \right] + \frac{\gamma\varepsilon}{\theta} \sum_{i=0}^{t} (1-\gamma\mu)^i
$$

$$
\leq (1-\gamma\mu)^{t+1} \left[ f(x^0) - f^* + \frac{\gamma}{2\theta} G^0 \right] + \frac{\varepsilon}{\theta\mu}.
$$

$\square$

**Lemma 12** (DIANA iteration). *Let Assumption 1, be satisfied and $\mathcal{C} \in \mathcal{D}_U(\omega, \varepsilon)$. Choose*

$$
0 < \gamma \leq \frac{1}{L \left( 1 + \sqrt{\frac{\rho + 2\omega^2\beta}{n}} \right)},
$$

*where $\beta = \rho = \frac{2\omega}{n}$. Define $V^t = \mathbb{E}\left[ f(x^t) + \frac{\gamma}{2}\|g^t - \nabla f(x^t)\|^2 + \frac{\gamma\beta\omega}{n}\frac{1}{n}\sum_{i=1}^{n}\|\nabla f_i(x^t) - h_i^t\|^2 \right]$.*
*Then, we have*

$$V^{t+1} \leq V^t - \frac{\gamma}{2}\mathbb{E}\|\nabla f(x^t)\|^2 + \frac{\gamma(1 + 2\omega\beta)}{2n}\varepsilon,$$

*and*

$$\frac{1}{T}\sum_{t=0}^{T-1}\mathbb{E}\|\nabla f(x^t)\|^2 \leq \frac{2V^0}{\gamma T} + \frac{1 + 2\beta\omega}{n}\varepsilon.$$

*Additionally, if Assumptions ?? and 2 are satisfied, $V^t = \mathbb{E}\Big[ f(x^t) - f^* + \frac{\gamma}{2}\|g^t - \nabla f(x^t)\|^2 +$*

*$\frac{\gamma\beta\omega}{n}\frac{1}{n}\sum_{i=1}^{n}\|\nabla f_i(x^t) - h_i^t\|^2\Big]$ and $\gamma$ is chosen as*

$$0 < \gamma \leq \min\left\{ \frac{1}{L\left(1 + \sqrt{\frac{2(\rho + 2\omega^2\beta)}{n}}\right)}, \frac{1}{4\omega\mu} \right\},$$

*then,*

$$V^{t+1} \leq (1 - \gamma\mu)V^t + \frac{\gamma(1 + 2\omega\beta)}{n}\varepsilon,$$

*and*

$$V^{t+1} \leq (1 - \gamma\mu)^{t+1}V^0 + \frac{\varepsilon(1 + 2\omega\beta)}{n\mu}.$$

*Proof.* From $L$-smoothness we obtain

$$
\begin{aligned}
f(x^{t+1}) &\leq f(x^t) + \langle \nabla f(x^t), x^{t+1} - x^t \rangle + \frac{L}{2}\|x^{t+1} - x^t\|^2 \\
&\leq f(x^t) - \gamma\langle \nabla f(x^t), g^t \rangle + \frac{L\gamma^2}{2}\|g^t\|^2 \\
&= f(x^t) - \frac{\gamma}{2}\|\nabla f(x^t)\|^2 + \left(\frac{L\gamma^2}{2} - \frac{\gamma}{2}\right)\|g^t\|^2 + \frac{\gamma}{2}\|g^t - \nabla f(x^t)\|^2.
\end{aligned}
$$

Now we analyze the term $E^{t+1} = \|g^{t+1} - \nabla f(x^{t+1})\|$:

$$
\begin{aligned}
\mathbb{E}_{\xi_t}\|g^{t+1} - \nabla f(x^t)\|^2 &= \mathbb{E}\left\| h^t + \frac{1}{n}\sum_{i=1}^{n}\mathcal{D}_U\left(\nabla f_i(x^{t+1}) - h_i^t\right) - \nabla f_i(x^t) \right\|^2 \\
&= \frac{1}{n^2}\sum_{i=1}^{n}\mathbb{E}_{\xi_t}\left\| h_i^t + \mathcal{D}_U\left(\nabla f_i(x^{t+1}) - h_i^t\right) - \nabla f_i(x^{t+1}) \right\|^2 \\
&\leq \frac{\omega - 1}{n^2}\sum_{i=1}^{n}\mathbb{E}_{\xi_t}\left\|\nabla f_i(x^{t+1}) - h_i^t\right\|^2 + \frac{\varepsilon}{n} \\
&\leq \frac{\omega - 1}{n^2}(1 + s)\sum_{i=1}^{n}\left\|\nabla f_i(x^t) - h_i^t\right\|^2 \\
&\quad + \frac{\omega - 1}{n}(1 + s^{-1})L^2\mathbb{E}_{\xi_t}\|x^{t+1} - x^t\|^2 + \frac{\varepsilon}{n} \\
&= \frac{\omega - 1}{n^2}(1 + s)\sum_{i=1}^{n}\left\|\nabla f_i(x^t) - h_i^t\right\|^2 \\
&\quad + \frac{\omega - 1}{n}(1 + s^{-1})L^2\gamma^2\mathbb{E}_{\xi_t}\|g^t\|^2 + +\frac{\varepsilon}{n}
\end{aligned}
$$

Take $s = 1$, then, after defining $H^t = \frac{1}{n}\sum_{i=1}^{n}\|\nabla f_i(x^t) - h_i^t\|^2$ we have

$$\mathbb{E}_{\xi_t}E^{t+1} \leq \frac{\beta}{n}H^t + \frac{\rho}{n}L^2\gamma^2\mathbb{E}_{\xi_t}\|g^t\|^2 + \frac{\varepsilon}{n},$$

where $\beta = \frac{2\omega}{n}$, $\rho = \frac{2\omega}{n}$. Now analyze the term $H^{t+1}$:

$$
\mathbb{E}_{\xi_t} \frac{1}{n} \sum_{i=1}^n \|\nabla f_i(x^{t+1}) - h_i^{t+1}\|^2 = \mathbb{E}_{\xi_t} \frac{1}{n} \sum_{i=1}^n \|\nabla f_i(x^{t+1}) - \alpha \mathcal{D}_U \left(\nabla f_i(x^{t+1}) - h_i^t\right) - h_i^t\|^2
$$

$$
\leq \left(1 - 2\alpha + \alpha^2 \omega\right) \frac{1}{n} \sum_{i=1}^n \mathbb{E}_{\xi_t} \|\nabla f_i(x^{t+1}) - h_i^t\|^2 + \varepsilon
$$

$$
\leq \left(1 - 2\alpha + \alpha^2 \omega\right)(1 + s) \frac{1}{n} \sum_{i=1}^n \|\nabla f_i(x^t) - h_i^t\|^2
$$

$$
+ \left(1 - 2\alpha + \alpha^2 \omega\right)(1 + s^{-1}) L^2 \mathbb{E}_{\xi_t} \|x^{t+1} - x^t\|^2 + \varepsilon
$$

$$
\leq \left(1 - 2\alpha + \alpha^2 \omega\right)(1 + s) \frac{1}{n} \sum_{i=1}^n \|\nabla f_i(x^t) - h_i^t\|^2
$$

$$
+ \left(1 - 2\alpha + \alpha^2 \omega\right)(1 + s^{-1}) L^2 \gamma^2 \mathbb{E}_{\xi_t} \|g^t\|^2 + \varepsilon
$$

Take $\alpha = \frac{1}{\omega}$ and $s = \frac{1}{2\omega}$. Then,

$$
\mathbb{E}_{\xi_t} H^{t+1} \leq \left(1 - \frac{1}{2\omega}\right) H^t + 2\omega L^2 \gamma^2 \mathbb{E}_{\xi_t} \|g^t\|^2 + \varepsilon.
$$

To combine these bounds we 1)add $E^{t+1}$ with multiplier $\frac{\gamma}{2}$ 2)add $H^{t+1}$ with multiplier $\frac{\gamma\beta\omega}{n}$ 3)take full expectation

$$
\mathbb{E}\left[f(x^{t+1}) + \frac{\gamma}{2} E^{t+1} + \frac{\gamma\beta\omega}{n} H^{t+1}\right] \leq \mathbb{E}f(x^t) - \frac{\gamma}{2} \mathbb{E}\|\nabla f(x^t)\|^2 + \frac{\gamma}{2} \mathbb{E}E^t
$$

$$
+ \left(\frac{\gamma\beta}{2n} + \frac{\gamma\beta\omega}{n}\left(1 - \frac{1}{2\omega}\right)\right) \mathbb{E}H^t +
$$

$$
+ \left(\frac{L\gamma^2}{2} - \frac{\gamma}{2} + \frac{\gamma^3 L^2(\rho + 2\omega^2\beta)}{2n}\right) \mathbb{E}\|g^t\|^2
$$

$$
+ \frac{\gamma(1 + 2\omega\beta)}{2n} \varepsilon
$$

$$
= \mathbb{E}f(x^t) - \frac{\gamma}{2} \mathbb{E}\|\nabla f(x^t)\|^2 + \frac{\gamma}{2} \mathbb{E}E^t + \frac{\gamma\beta\omega}{n} \mathbb{E}H^t +
$$

$$
+ \left(\frac{L\gamma^2}{2} - \frac{\gamma}{2} + \frac{\gamma^3 L^2(\rho + 2\omega^2\beta)}{2n}\right) \mathbb{E}\|g^t\|^2
$$

$$
+ \frac{\gamma(1 + 2\omega\beta)}{2n} \varepsilon.
$$

After the choice of $\gamma$ we have

$$
\mathbb{E}\left[f(x^{t+1}) + \frac{\gamma}{2} E^{t+1} + \frac{\gamma\beta\omega}{n} H^{t+1}\right] \leq \mathbb{E}f(x^t) - \frac{\gamma}{2} \mathbb{E}\|\nabla f(x^t)\|^2 + \frac{\gamma}{2} \mathbb{E}E^t + \frac{\gamma\beta\omega}{n} \mathbb{E}H^t
$$

$$
+ \frac{\gamma(1 + 2\omega\beta)}{2n} \varepsilon.
$$

Therefore we get

$$
\frac{1}{T} \sum_{t=1}^T \mathbb{E}\left\|\nabla f(x^t)\right\|^2 \leq \frac{2f(x^0) + \gamma E^0 + \frac{2\gamma\beta\omega}{n} H^0}{\gamma T} + \frac{1 + 2\beta\omega}{n} \varepsilon.
$$

In the PL case we add $E^{t+1}$ with multiplier $\gamma$, $H^{t+1}$ with multiplier $\frac{2\gamma\beta\omega}{n}$ and extract $f^*$:

$$
\begin{aligned}
\mathbb{E}\left[f(x^{t+1}) - f^* + \gamma E^{t+1} + \frac{2\gamma\beta\omega}{n}H^{t+1}\right] \leq\ & \mathbb{E}f(x^t) - f^* - \frac{\gamma}{2}\mathbb{E}\|\nabla f(x^t)\|^2 + \frac{\gamma}{2}\mathbb{E}E^t \\
& + \left(\frac{\gamma\beta}{2n} + \frac{2\gamma\beta\omega}{n}\left(1 - \frac{1}{2\omega}\right)\right)\mathbb{E}H^t + \\
& + \left(\frac{L\gamma^2}{2} - \frac{\gamma}{2} + \frac{\gamma^3 L^2(\rho + 2\omega^2\beta)}{n}\right)\mathbb{E}\|g^t\|^2 \\
& + \frac{\gamma(1 + 2\omega\beta)}{n}\varepsilon \\
\leq\ & (1 - \gamma\mu)\left(f(x^t) - f^*\right) + \frac{\gamma}{2}\mathbb{E}E^t \\
& + \left(1 - \frac{1}{4\omega}\right)\frac{2\gamma\beta\omega}{n}\mathbb{E}H^t \\
& + \left(\frac{L\gamma^2}{2} - \frac{\gamma}{2} + \frac{\gamma^3 L^2(\rho + 2\omega^2\beta)}{n}\right)\mathbb{E}\|g^t\|^2 \\
& + \frac{\gamma(1 + 2\omega\beta)}{n}\varepsilon.
\end{aligned}
$$

After the choice of $\gamma$:

$$
\mathbb{E}\left[f(x^{t+1}) + \gamma E^{t+1} + \frac{2\gamma\beta\omega}{n}H^{t+1}\right] \leq (1 - \gamma\mu)\mathbb{E}\left(f(x^t) + \gamma E^t + \frac{2\gamma\beta\omega}{n}H^t\right) + \frac{\gamma(1 + 2\omega\beta)}{n}\varepsilon
$$

After unrolling the sequence we obtain

$$
\begin{aligned}
\mathbb{E}\left[f(x^{t+1}) + \gamma E^{t+1} + \frac{2\gamma\beta\omega}{n}H^0\right] \leq\ & (1 - \gamma\mu)^{t+1}\left[f(x^0) + \gamma E^0 + \frac{2\gamma\beta\omega}{n}H^0\right] \\
& + \frac{\gamma(1 + 2\omega\beta)\varepsilon}{n}\sum_{i=0}^{t}(1 - \gamma\mu)^i \\
\leq\ & (1 - \gamma\mu)^{t+1}\left[f(x^0) + \gamma E^0 + \frac{2\gamma\beta\omega}{n}H^0\right] \\
& + \frac{(1 + 2\omega\beta)\varepsilon}{\mu n}.
\end{aligned}
$$

$\square$

**Lemma 13** (DASHA iteration). *Let Assumption 1, be satisfied and $\mathcal{C} \in \mathcal{D}_U(\omega, \varepsilon)$. Choose $m = \frac{1}{2\omega+1}$ and*

$$
0 < \gamma \leq \frac{1}{L\left(1 + \sqrt{\frac{16\omega(2\omega+1)}{n}}\right)}.
$$

*Define $V^t = \mathbb{E}\left[f(x^{t+1}) + \gamma(2\omega + 1)E^{t+1} + \frac{2\gamma\omega}{n}H^{t+1}\right]$. Then, we have*

$$
V^{t+1} \leq V^t - \frac{\gamma}{2}\mathbb{E}\|\nabla f(x^t)\|^2 + \frac{\gamma(1 + 4\omega)}{n}\varepsilon,
$$

*and*

$$
\frac{1}{T}\sum_{t=0}^{T-1}\mathbb{E}\|\nabla f(x^t)\|^2 \leq \frac{2V^0}{\gamma T} + \frac{2(1 + 4\omega)}{n}\varepsilon.
$$

*Additionally, if Assumptions ?? and 2 are satisfied, $V^t = \mathbb{E}\left[f(x^{t+1}) - f^* + \gamma(2\omega + 1)E^{t+1} + \frac{2\gamma\omega}{n}H^{t+1}\right]$ and $\gamma$ is chosen as*

$$
0 < \gamma \leq \min\left\{\frac{1}{L\left(1 + \sqrt{\frac{40\omega(2\omega+1)}{n}}\right)}, \frac{1}{2(2\omega + 1)\mu}\right\},
$$

*then,*

$$V^{t+1} \leq (1 - \gamma\mu)V^t + \frac{\gamma(1 + 10\omega)}{n}\varepsilon,$$

*and*

$$V^{t+1} \leq (1 - \gamma\mu)^{t+1}V^0 + \frac{\varepsilon(1 + 10\omega)}{n\mu}.$$

*Proof.* From $L$-smoothness we obtain

$$
\begin{aligned}
f(x^{t+1}) &\leq f(x^t) + \langle \nabla f(x^t), x^{t+1} - x^t \rangle + \frac{L}{2}\|x^{t+1} - x^t\|^2 \\
&\leq f(x^t) - \gamma\langle \nabla f(x^t), g^t \rangle + \frac{L\gamma^2}{2}\|g^t\|^2 \\
&= f(x^t) - \frac{\gamma}{2}\|\nabla f(x^t)\|^2 + \left(\frac{L\gamma^2}{2} - \frac{\gamma}{2}\right)\|g^t\|^2 + \frac{\gamma}{2}\|g^t - \nabla f(x^t)\|^2
\end{aligned}
$$

Now we analyze the term $E^{t+1} = \|g^{t+1} - \nabla f(x^{t+1})\|^2$. Taking the conditional expectation $\mathbb{E}_{\xi_t} = \mathbb{E}\left[\cdot \mid x^1, \ldots, x^t\right]$, we have

$$
\mathbb{E}_{\xi_t}\|g^{t+1} - \nabla f(x^{t+1})\|^2 = \mathbb{E}_{\xi_t}\Big\|g^t + \frac{1}{n}\sum_{i=1}^{n}\mathcal{D}_U\big(\nabla f_i(x^{t+1}) - \nabla f_i(x^t)
$$

$$
- \quad m(g_i^t - \nabla f_i(x^t))) - \nabla f(x^{t+1})\Big\|^2
$$

$$
= \quad \mathbb{E}_{\xi_t}\Big\|\frac{1}{n}\sum_{i=1}^{n}\mathcal{D}_U\big(\nabla f_i(x^{t+1}) - \nabla f_i(x^t) - m(g_i^t - \nabla f_i(x^t))\big)
$$

$$
- \quad \frac{1}{n}\sum_{i=1}^{n}\nabla f_i(x^{t+1}) - \nabla f_i(x^t) - m(g_i^t - \nabla f_i(x^t))\Big\|^2 + (1 - m)^2\|g^t - \nabla f(x^t)\|^2
$$

$$
= \quad \frac{1}{n^2}\sum_{i=1}^{n}\mathbb{E}_{\xi_t}\Big\|\mathcal{D}_U\big(\nabla f_i(x^{t+1}) - \nabla f_i(x^t) - m(g_i^t - \nabla f_i(x^t))\big)
$$

$$
- \quad (\nabla f_i(x^{t+1}) - \nabla f_i(x^t) - m(g_i^t - \nabla f_i(x^t))\Big\|^2 + (1 - m)^2\|g^t - \nabla f(x^t)\|^2
$$

$$
\leq \quad \frac{\omega}{n^2}\sum_{i=1}^{n}\|\nabla f_i(x^{t+1}) - \nabla f_i(x^t) - m(g_i^t - \nabla f_i(x^t))\|^2 + (1 - m)^2\|g^t - \nabla f(x^t)\|^2 + \frac{\varepsilon}{n}.
$$

Use Young's inequality:

$$
\begin{aligned}
\mathbb{E}_{\xi_t}\|g^{t+1} - \nabla f(x^{t+1})\|^2 &\leq \frac{2\omega}{n^2}\sum_{i=1}^{n}\mathbb{E}_{\xi_t}\|\nabla f_i(x^{t+1}) - \nabla f_i(x^t)\|^2 \\
&\quad + \frac{2m^2\omega}{n}\sum_{i=1}^{n}\|g_i^t - \nabla f_i(x^t)\|^2 + (1 - m)^2\|g^t - \nabla f(x^t)\|^2 + \frac{\varepsilon}{n} \\
&\leq \frac{2\omega L^2}{n}\mathbb{E}_{\xi_t}\|x^{t+1} - x^t\|^2 + \frac{2m^2\omega}{n}\sum_{i=1}^{n}\|g_i^t - \nabla f_i(x^t)\|^2 \\
&\quad + (1 - m)^2\|g^t - \nabla f(x^t)\|^2 + \frac{\varepsilon}{n} \\
&= \frac{2\omega L^2\gamma^2}{n}\mathbb{E}_{\xi_t}\|g^t\|^2 + \frac{2m^2\omega}{n}\sum_{i=1}^{n}\|g_i^t - \nabla f_i(x^t)\|^2 \\
&\quad + (1 - m)^2\|g^t - \nabla f(x^t)\|^2 + \frac{\varepsilon}{n}.
\end{aligned}
$$

Now analyze the term $H^{t+1} = \frac{1}{n}\sum\limits_{i=1}^{n}\|g_i^{t+1} - \nabla f_i(x^{t+1})\|^2$:

$$\mathbb{E}_{\xi_t}\frac{1}{n}\sum_{i=1}^{n}\|g_i^{t+1} - \nabla f_i(x^{t+1})\|^2 = \mathbb{E}_{\xi_t}\frac{1}{n}\sum_{i=1}^{n}\|g_i^t + \mathcal{D}_U(\nabla f_i(x^{t+1}) - \nabla f_i(x^t)$$

$$- \quad m(g_i^t - \nabla f_i(x^t))) - \nabla f_i(x^{t+1})\|^2$$

$$= \quad \mathbb{E}_{\xi_t}\frac{1}{n}\sum_{i=1}^{n}\|g_i^{t+1} - \nabla f_i(x^{t+1})\|^2 = \mathbb{E}_{\xi_t}\frac{1}{n}\sum_{i=1}^{n}\|g_i^t + \mathcal{D}_U(\nabla f_i(x^{t+1}) - \nabla f_i(x^t)$$

$$- \quad m(g_i^t - \nabla f_i(x^t))) - \nabla f_i(x^{t+1}) - \nabla f_i(x^t) - m(g_i^t - \nabla f_i(x^t))\|^2$$

$$+ \quad (1-m)^2\|g_i^t - \nabla f_i(x^t)\|^2$$

$$\leq \quad \mathbb{E}_{\xi_t}\frac{1}{n}\sum_{i=1}^{n}\omega\|\nabla f_i(x^{t+1}) - \nabla f_i(x^t) - m(g_i^t - \nabla f_i(x^t))\|^2 + (1-m)^2\|g_i^t - \nabla f_i(x^t)\|^2 + \varepsilon$$

$$\leq \quad 2\omega L^2\mathbb{E}_{\xi_t}\|x^{t+1} - x^t\|^2 + \left(2m^2\omega + (1-m^2)\right)\sum_{i=1}^{n}\|g_i^t - \nabla f_i(x^t)\|^2 + \varepsilon$$

$$\leq \quad 2\omega L^2\gamma^2\mathbb{E}_{\xi_t}\|g^t\|^2 + \left(2m^2\omega + (1-m^2)\right)\sum_{i=1}^{n}\|g_i^t - \nabla f_i(x^t)\|^2 + \varepsilon$$

Take $m = \frac{1}{2\omega+1}$, add $E^{t+1}$ with multiplier $\frac{\gamma}{m}$, $H^{t+1}$ with multiplier $\frac{2\gamma\omega}{n}$ and take full expectation, then

$$\mathbb{E}\left[f(x^{t+1}) + \gamma(2\omega+1)E^{t+1} + \frac{2\gamma\omega}{n}H^{t+1}\right] \leq \mathbb{E}\Big[f(x^t) + \gamma(2\omega+1)\|g^t - \nabla f(x^t)\|^2 E^t$$

$$+ \quad \frac{2\gamma\omega}{n}H^t\Big] - \frac{\gamma}{2}\mathbb{E}\|\nabla f(x^t)\|^2 + \left(\frac{L\gamma^2}{2} - \frac{\gamma}{2} + \frac{\gamma^3 L^2 2\omega(2\omega+1)}{n} + \frac{\gamma^3 L^2 4\omega^2}{n}\right)\mathbb{E}\|g^t\|^2$$

$$+ \quad \frac{\gamma(4\omega+1)}{n}\varepsilon.$$

After the choice of $\gamma$:

$$\mathbb{E}\left[f(x^{t+1}) + \gamma(2\omega+1)E^{t+1} + \frac{2\gamma\omega}{n}H^{t+1}\right] \leq \mathbb{E}\Big[f(x^t) + \gamma(2\omega+1)\|g^t - \nabla f(x^t)\|^2 E^t$$

$$+ \quad \frac{2\gamma\omega}{n}H^t\Big] - \frac{\gamma}{2}\mathbb{E}\|\nabla f(x^t)\|^2 + \frac{\gamma(4\omega+1)}{n}\varepsilon.$$

Therefore, we get

$$\frac{1}{T}\sum_{t=1}^{T}\mathbb{E}\|\nabla f(x^t)\|^2 \leq \frac{2f(x^0) + \frac{2\gamma}{m}E^0 + \frac{4\gamma\omega}{n}H^0}{\gamma T} + \frac{8\omega+2}{n}\varepsilon$$

In the PL case we add $E^{t+1}$ with multiplier $\frac{2\gamma}{m}$, $H^{t+1}$ with multiplier $\frac{8\gamma\omega}{n}$ and extract $f^*$, Then, with the proper choice of $\gamma$

$$\mathbb{E}\left[f(x^{t+1}) - f^* + \gamma(2\omega+1)E^{t+1} + \frac{8\gamma\omega}{n}H^{t+1}\right]$$

$$\leq \quad (1-\gamma\mu)\mathbb{E}\left[f(x^t) - f^* + \gamma(2\omega+1)E^t + \frac{8\gamma\omega}{n}H^t\right] + \frac{\gamma(10\omega+1)}{n}\varepsilon$$

After unrolling the sequence we obtain

$$\mathbb{E}\left[f(x^{t+1}) - f^* + \gamma(2\omega+1)E^{t+1} + \frac{8\gamma\omega}{n}H^{t+1}\right]$$

$$\leq \quad (1-\gamma\mu)^{t+1}\left[f(x^{t0}) - f^* + \gamma(2\omega+1)E^0 + \frac{8\gamma\omega}{n}H^0\right]$$

$$+ \quad \frac{\gamma(1+10\omega)\varepsilon}{n}\sum_{i=0}^{t}(1-\gamma\mu)^i$$

$$\leq \quad (1-\gamma\mu)^{t+1}\left[f(x^0) - f^* + \gamma(2\omega+1)E^0 + \frac{8\gamma\omega}{n}H^0\right] + \frac{(1+10\omega)\varepsilon}{\mu n}.$$

## C  DYNAMIC DATA TYPE

**Theorem 3** (Nonconvex case). *Let $V^t$ be a Lyapunov function, satisfying*
$$V^{t+1} \leq V^t - \frac{\gamma}{2}\|\nabla f(x^t)\|^2 + \delta(\mathcal{A})\gamma.$$
*Let us update the rounding grid every $N$ iterations. Let $\delta_0$ be the approximation error for the first grid. Then, we achieve following exactness in gradients' norm:*
$$\frac{1}{T}\sum_{t=1}^{T-1}\|\nabla f(x^t)\|^2 \leq \frac{2V^0}{\gamma T} + \frac{N\delta_0}{T(1-\rho)},$$
*where $\rho = \delta(\mathcal{A}_{t+1})/\delta(\mathcal{A}_t)$*

*Proof.* Consider, that we update the rounding at iterations $N_i$, then:
$$V^{N_k} - V^0 = \sum_{i=1}^{N_k-1} V^{i+1} - V^i = \sum_{t=1}^{k}\sum_{i=N^t}^{N^{t+1}-1} V^{i+1} - V^i = \sum_{t=1}^{k}\sum_{i=N^t}^{N^{t+1}-1} -\frac{\gamma}{2}\|\nabla f(x^i)\|^2 + \delta_t\gamma$$
$$\leq -\frac{\gamma}{2}\sum_{i=1}^{N^k-1}\|\nabla f(x^i)\|^2 + \gamma\sum_{t=0}^{k}(N_{t+1}-N_t)\delta_t.$$
Hence, after defining $N_t^* = N_{t+1} - N_t$, we have
$$\frac{1}{N_k}\sum_{i=0}^{N_k-1}\|\nabla f(x^i)\|^2 \leq \frac{2V^0}{\gamma N_k} + \frac{1}{N_k}\sum_{t=0}^{k}N_t^*\delta_t.$$
If $\delta_t = \delta_0 \cdot \rho^t$, where $0 < \rho < 1$, which, for example, is true for grids, that renews by dividing all their values by $1/\sqrt{\rho}$. Also, take $N_t^* \equiv N$, then
$$\frac{1}{N_k}\sum_{i=0}^{N_k-1}\|\nabla f(x^i)\|^2 \leq \frac{2V^0}{\gamma kN} + \frac{\sum_{t=1}^{k}\delta_0 N\alpha^t}{kN} \leq \frac{2V^0}{\gamma kN} + \frac{\delta_0}{k(1-\rho)}.$$
$\qquad\qquad\qquad\qquad\qquad\qquad\qquad\qquad\qquad\qquad\qquad\qquad\qquad\qquad\qquad\qquad\square$

**Theorem 4** (PL case). *Let $V^t$ be a Lyapunov function, satisfying*
$$V^{t+1} \leq (1-\gamma\mu)V^t + \gamma\delta(\mathcal{A}.)$$
*Let $\rho = \delta(\mathcal{A}_{t+1})/\delta(\mathcal{A}_t)$ and the maximum element of $\mathcal{A}_0$ be $a_{max}$. If $\sqrt{\rho} > \frac{\delta(\mathcal{A}_0)}{\mu a_{max}}$, then after updating the grid every*
$$N = \frac{\log\left(\sqrt{\rho} - \frac{\delta(\mathcal{A}_0)}{\mu a_{max}}\right)}{\log(1-\gamma\mu)}$$
*we will achieve accuracy $\varepsilon$ in $\mathcal{O}\left(\log\frac{1}{\varepsilon}\right)$ iterations.*

*Proof.* As compressors introduced earlier satisfy assumptions on (inexact) (un)biasedness only, when the maximum compressed element do not exceed the maximum element of the grid, we are going to obtain the number of iterations needed, to satisfy this. Consider grids, that updates by dividing all its elements by $1/\sqrt{\rho}$. We will prove the needed statement by induction. We have
$$V^{N_i} \leq \rho^{\frac{i}{2}} a_{max},$$
where $N_i$ is the iteration we are renewing the rounding lattice and $a_{max}$ is the maximum element. Then, if $N_i^* = N_{i+1} - N_i$, we have
$$V^{N_{i+1}} \leq (1-\gamma\mu)^{N_i^*} V^{N_i} + \gamma\rho^i\delta_0 \sum_{k=0}^{N_i^*-1}(1-\gamma\mu)^k$$
$$\leq (1-\gamma\mu)^{N_i^*}\rho^{i/2} a_{max} + \gamma\rho^i\delta_0 \sum_{k=0}^{N_i^*-1}(1-\gamma\mu)^k$$
$$\leq (1-\gamma\mu)^{N_i^*}\rho^{i/2} a_{max} + \frac{\rho^i\delta_0}{\mu}$$

To get $V^{N_{i+1}} \leq \rho^{\frac{i+1}{2}} a_{max}$, we need

$$(1 - \gamma\mu)^{N_i^*} \leq \sqrt{\rho} - \frac{\rho^{i/2}\delta_0}{\mu a_{max}}.$$

It is sufficient to satisfy

$$(1 - \gamma\mu)^{N_i^*} \leq \sqrt{\rho} - \frac{\delta_0}{\mu a_{max}}.$$

Hence, we need to take restarts every

$$N_i^* \geq N = \frac{\log\left(\sqrt{\rho} - \frac{\delta_0}{\mu a_{max}}\right)}{\log(1 - \gamma\mu)}.$$

As a result, needed accuracy will be achieved in

$$k \geq 2\frac{\log\frac{\varepsilon}{a_{max}}}{\log\rho}$$

$\square$

# D DECLARATION OF LLM USAGE

We employed Large Language Models to improve the clarity and style of the text.

