# OpenReview forum: "Dynamic Compression in Distributed Communications for Reduction of Transmission Losses"
_ICLR.cc/2026/Conference — Submitted to ICLR 2026_

### Official Review · Reviewer_WzCT · 2025-10-27

**Soundness:** 2
**Presentation:** 2
**Contribution:** 3
**Rating:** 4
**Confidence:** 3

**Summary:**

This paper identifies a limitation in existing theoretical frameworks for compression operators when applying to low-bit quantization. To address this, the authors propose new operator classes ($\mathbb{U}(\omega, \epsilon)$ and $\mathbb{B}^3(\alpha, \epsilon)$) that explicitly include this absolute error term, $\epsilon$. The paper then introduces a "Dynamic Data Type" method, which adaptively refines the quantization grid to make $\epsilon$ diminish during training, thereby enabling convergence to an exact solution instead of a residual error neighborhood. The theoretical analysis is applied to state-of-the-art algorithms like EF21, DIANA, and DASHA.

**Strengths:**

The paper identifies a practical limitation in the standard theoretical framework for compression operators. The observation that fixed-grid, low-bit quantization introduces a non-diminishing absolute error $\epsilon$, which is ignored by traditional relative-error definitions, is a valid and inspiring contribution. Besides, the theoretical analysis appears to be solid.

**Weaknesses:**

W1, The paper suffers from significant readability issues that hinder understanding. The main experimental plots on page 8 lack proper "Figure" numbers and captions. Crucial new notation, specifically the core concepts of the $\mathbb{B}^3(\alpha, \epsilon)$ and $\mathbb{U}(\omega, \epsilon)$ operator classes, are introduced implicitly rather than through formal definitions.

W2, The experimental validation feels thin and, in some cases, confusing. The theory is developed for EF21, DIANA, and DASHA, but the experiments appear to focus only on EF21. The analysis of the results is brief, and some of the experimental plots are confusing and seem to contradict the paper's own claims (as detailed in Question 2).

**Questions:**

Q1 Regarding the main convergence results, could the authors elaborate on how the absolute error term $\epsilon$ precisely impacts the convergence guarantees? Furthermore, does this new theoretical framework gracefully cover traditional (non-low-bit) biased and unbiased compressors?

Q2 In the ResNet gradient norm plots (page 8, right column), there are two major concerns. First, why do the 'dynamic' (top row) and 'static' (bottom row) experiments have different Y-axis starting points (e.g., $10^3$ vs. $approx 10^1$)? Besides, the plots seem to show that the static baseline (bottom row) actually converges to a *lower* gradient norm (approx. $10^{-4}$) than the proposed dynamic method (approx. $10^{-2}$). Can the authors please explain this discrepancy?

Q3 How does the proposed dynamic quantization grid algorithm perform when applied to the other compressors analyzed in the theory, such as DIANA and DASHA? The current experiments seem limited to EF21.

---

### Official Review · Reviewer_yMD1 · 2025-10-30

**Soundness:** 3
**Presentation:** 2
**Contribution:** 2
**Rating:** 2
**Confidence:** 3

**Summary:**

The paper addresses the communication bottleneck in distributed and federated learning by introducing a novel framework for dynamic compression. Traditional compression operators rely on fixed variance assumptions and do not adapt as gradients shrink during training, leading to inefficiencies. To overcome this, the authors propose new definitions for biased and unbiased compression operators that incorporate finite-grid rounding errors, and introduce dynamic data types that adjust compression precision throughout the optimization process. The approach is theoretically analyzed and integrated into state-of-the-art distributed optimization algorithms such as EF21, DIANA, and DASHA, with convergence guarantees under both nonconvex and PL conditions. Experimental results on logistic regression and ResNet tasks demonstrate that dynamic data types significantly reduce communication costs while maintaining or improving convergence speed.

**Strengths:**

1.	The idea of dynamic compression (adjusting accuracy during the training phase) is of practical significance and solves the problem that static quantization cannot accurately control errors.
2.	The authors derive a new class of compression operators and rigorously prove their compatibility and convergence with existing operators.

**Weaknesses:**

1.	Limited Experimental Scale: The experiments primarily focused on logistic regression and small- to medium-scale ResNet image classification tasks. They were not validated in scenarios such as large-scale distributed training (e.g., ImageNet and GPT-like tasks), heterogeneous networks with varying bandwidths, or large numbers of nodes.
2.	Although the paper compares the three frameworks of EF21, DIANA, and DASHA, these frameworks all come from the same research school and lack comparison with other dynamic compression algorithms in recent years.
3.	Theoretical convergence analysis relies heavily on assumptions and lacks rigorous proofs for non-convex complex models.

**Questions:**

1.	There are problems with the structure of the article. It is best not to have definitions and subsections in the Introduction. You can consider putting section 1.1 in Preliminaries. Meanwhile, there are grammatical and typographical issues. It also misses Conclusion chapter and the caption of the figures.
2.	Ablation experiments are missing: The paper does not systematically study: the individual effects of different 𝑝 on the convergence rate; the performance of dynamic compression under different gradient distributions (sparse/dense); and whether it can be used in conjunction with Top-k sparsification.

---

### Official Review · Reviewer_fp2V · 2025-10-30

**Soundness:** 1
**Presentation:** 1
**Contribution:** 1
**Rating:** 2
**Confidence:** 4

**Summary:**

The paper studies compression operators for distributed learning, where communication between clients and the central server is often the main bottleneck. The proposed operators compress transmitted values (such as gradients) by rounding each coordinate to a finite grid. Two variants are considered: an unbiased one that uses probabilistic rounding based on proximity to grid points, and a biased one that rounds to the nearest grid point.
The key idea is the use of a dynamic grid that adapts over time. As the communicated values (e.g., gradients) become smaller during training, the grid range is reduced, improving precision and communication efficiency.

**Strengths:**

The idea of using a dynamic grid for quantization is reasonable and intuitive, though I am not sure whether it is novel or has been studied before.

**Weaknesses:**

1. **Poor writing quality**
   The paper is poorly written, with many grammatical errors and unclear sentences. Several key terms are never defined, e.g., $\mathbb{B}^3(\alpha, \varepsilon)$ and $\mathbb{U}(w, \varepsilon)$, even though they seem central to the main results. Other notations, such as $\varepsilon(\mathcal{A})$, also appear without explanation. Overall, the presentation feels rushed and unpolished.

2. **Unclear implementation of the dynamic grid**
   The update of the grid depends on a condition called *Degree\_cond*, which is said to rely on some threshold of the communicated values. However, the paper does not explain how this threshold should be chosen.

3. **Lack of theoretical analysis for the dynamic scheme**
   There is no theoretical justification for how the dynamically changing quantization set should be updated or how change strategies affect convergence.

4. **Limited practical relevance**
   In deep learning tasks, the gradient norm often does not converge to zero, so the proposed dynamic-grid approach may not be practical in real-world training scenarios.

**Questions:**

See weaknesses.

---

### Official Review · Reviewer_2QdW · 2025-10-31

**Soundness:** 1
**Presentation:** 2
**Contribution:** 2
**Rating:** 2
**Confidence:** 4

**Summary:**

This paper proposes Dynamic Compression in Distributed Communications — a framework that introduces adaptive (dynamic) compression operators and “dynamic data types” for distributed optimization. The main idea is to adjust the quantization or rounding grid during training so that compression precision increases as gradients become smaller, thereby reducing transmission costs without losing convergence guarantees. The authors formalize new definitions for biased and unbiased compressors that consider finite quantization grids and additive error terms, analyze their convergence under smoothness and PL conditions, and integrate them into existing algorithms such as EF21, DIANA, and DASHA. Experiments on logistic regression (LibSVM datasets) and ResNet image classification show that dynamic compression improves communication efficiency compared to static compression.

**Strengths:**

S1 The theoretical sections are detailed and mathematically rigorous, with convergence analyses for several classes of compressors.

S2 The proposed “dynamic data type” idea is conceptually simple and appealing — gradually increasing precision as training progresses aligns with observed gradient magnitude decay.

S3 The proposed approach is applied to several algorithms.

**Weaknesses:**

W1 Insufficient experimental evaluation. The experiments are very limited — mostly small-scale logistic regression (mushrooms, a9a, w8a) and a single ResNet test. There are no large-scale distributed benchmarks, no results on real multi-node setups, and no comparisons with recent adaptive or learned compression methods (e.g., natural compression, adaptive quantization, or low-bit dynamic quantizers).

W2 Lack of ablation studies. There is no analysis on how parameters like grid update frequency, percentile choice, or rounding base
𝑝 affect convergence or communication cost. The “dynamic data type” mechanism is treated as a black box, without studying its sensitivity or stability.

W3 Missing important comparisons. There's are many related methods in the literature but the paper does not have any comparison with other methods.

W4 The paper is math-heavy with unclear separation between motivation, theoretical contribution, and implementation. The experimental section is particularly brief and lacks reproducibility details (e.g., hardware, network topology, communication budget).

**Questions:**

See above

---

### Official Review · Reviewer_6MJr · 2025-11-01

**Soundness:** 2
**Presentation:** 1
**Contribution:** 2
**Rating:** 2
**Confidence:** 3

**Summary:**

The paper proposes dynamic data type–based compressors for distributed/federated optimization. Instead of assuming infinite-precision lattices, the authors formalize finite-grid rounding (both unbiased and biased variants) and integrate them into EF21, DIANA, and DASHA, with convergence analyses that exploit geometric grid refinement. Experiments on logistic regression (LibSVM) and ResNet show reduced communication for comparable accuracy.

**Strengths:**

+ The paper identifies a gap in classic compressor analyses (infinite grids/second-moment focus) and introduces finite-grid rounding operators with additive-error terms and precise definitions for unbiased/biased rounding.

+ With a geometric refinement schedule, the analysis achieves standard O(1/K) nonconvex rates without residual and a linear rate under PL-type conditions, directly tying accuracy to grid tightening.

+ The experiments show practical integration and positive results.

**Weaknesses:**

- No conclusion part is involved in this paper.

- Dynamic grids imply control-plane costs. However, these implementation/latency costs aren’t quantified, so real-world gains may be overstated versus “bits on paper.”

- There’s no ablation on alternative schedules or robustness of the trigger, nor prescriptive guidance for combining with other biased/unbiased compressors beyond basic closure statements.

**Questions:**

+ Can you follow the ICLR format with providing a conclusion in this paper?

+ How sensitive are results to the refinement factor?

---

### Meta-Review · Area_Chair_fXaJ · 2026-01-01

**Summary:**

The paper received consistently negative review comments. The major weaknesses of the paper are summarized as follows:
1. Poor writing and presentation
2. Limited experimental scope and comparison
3. Lack of ablation studies and sensitivity analysis

**Reviewer Concerns:**

There is no rebuttal submitted by the authors.

**Reviewer Scores:**

There is no discussion between authors and reviewers.

---

### Decision · Program_Chairs · 2026-01-26

Reject